# Pan-Cancer Analysis Identifies *MNX1* and Associated Antisense Transcripts as Biomarkers for Cancer

**DOI:** 10.3390/cells11223577

**Published:** 2022-11-11

**Authors:** Denise Ragusa, Sabrina Tosi, Cristina Sisu

**Affiliations:** 1Leukaemia and Chromosome Research Laboratory, College of Health, Medicine and Life Sciences, Brunel University London, Kingston Lane, Uxbridge UB8 3PH, UK; 2Centre for Genome Engineering and Maintenance (CenGEM), College of Health, Medicine and Life Sciences, Brunel University London, Kingston Lane, Uxbridge UB8 3PH, UK; 3Department of Life Sciences, College of Health, Medicine and Life Sciences, Brunel University London, Kingston Lane, Uxbridge UB8 3PH, UK

**Keywords:** transcription, pan-cancer, *MNX1*, biomarker, diagnosis, prognosis, cancer cell development

## Abstract

The identification of diagnostic and prognostic biomarkers is a major objective in improving clinical outcomes in cancer, which has been facilitated by the availability of high-throughput gene expression data. A growing interest in non-coding genomic regions has identified dysregulation of long non-coding RNAs (lncRNAs) in several malignancies, suggesting a potential use as biomarkers. In this study, we leveraged data from large-scale sequencing projects to uncover the expression patterns of the *MNX1* gene and its associated lncRNAs *MNX1*-*AS1* and *MNX1*-*AS2* in solid tumours. Despite many reports describing *MNX1* overexpression in several cancers, limited studies exist on *MNX1*-*AS1* and *MNX1*-*AS2* and their potential as biomarkers. By employing clustering methods to visualise multi-gene relationships, we identified a discriminative power of the three genes in distinguishing tumour vs. normal samples in several cancers of the gastrointestinal tract and reproductive systems, as well as in discerning oesophageal and testicular cancer histological subtypes. Notably, the expressions of *MNX1* and its antisenses also correlated with clinical features and endpoints, uncovering previously unreported associations. This work highlights the advantages of using combinatory expression patterns of non-coding transcripts of differentially expressed genes as clinical evaluators and identifies *MNX1*, *MNX1*-*AS1*, and *MNX1*-*AS2* expressions as robust candidate biomarkers for clinical applications.

## 1. Introduction

Cancer represents a major contributor of morbidity and mortality worldwide, with current estimates suggesting that one in two people in the UK will develop cancer in their lifetime [1,2]. Despite significant improvements in therapeutic interventions and clinical outcomes in recent years, accurate patient diagnoses are imperative for maximising clinical success. Thus, the identification of genes as diagnostic and prognostic markers is a major objective in cancer research. The availability of large genome-wide studies is now facilitating the discovery of clinically relevant genes [3,4].

Motor neuron and pancreas homeobox protein 1 (*MNX1*), also known as *HLXB9*, is a key developmental homeobox gene coding for the transcription factor HB9. *MNX1* is involved in the differentiation and development of pancreatic and neuronal cells, and commonly expressed in these respective tissues [5,6,7,8]. The past decade saw a plethora of studies highlighting the dysregulation of *MNX1* in various cancers, including leukaemia [9], lymphoma [10], cancers of the breast [11,12], prostate [13], bladder [14], colorectal [15], brain [16,17], hepatocellular carcinomas [18], and pancreatic tumours [19]. However, the function and activity of *MNX1* in cancer is yet to be elucidated.

At the chromosomal locus 7q36.3, the *MNX1* gene is associated with two long non-coding RNAs (lncRNA)—the antisense transcripts *MNX1*-*AS1* (long intergenic non-coding RNA; lincRNA) and *MNX1*-*AS2* (antisense lncRNA). *MNX1* and *MNX1*-*AS1* share a promoter region, while *MNX1*-*AS2* is under the control of a second intergenic promoter (Figure 1A), however their transcriptional regulation remains elusive. Earlier studies have associated *MNX1*-*AS1* with the pathogenesis of a wide range of malignancies (e.g., bladder, breast, lung, prostate, oesophageal, gastrointestinal (GI), and gynaecological) in virtue of its regulatory activity on modulating malignant processes of proliferation, apoptosis, invasion/metastases, and migration (reviewed in [20]). The overexpression of *MNX1*-*AS1* has also been correlated with dismal clinical outcomes, such as shorter overall survival (OS), advanced clinical stage, and TNM classifiers [21]. By contrast, to date there are no reports on the function and activity of *MNX1*-*AS2* in cancer.

In this study, we leveraged the large-scale data from the Cancer Genome Atlas (TCGA) and Genotype Tissue Expression (GTEx) consortia projects to explore the transcriptional activity and function of the *MNX1* gene and its associated lncRNAs *MNX1*-*AS1* and *MNX1*-*AS2* in cancer and to uncover their clinical potential as biomarkers.

## 2. Methods

### 2.1. Data Availability, Sample Selection and Expression Analysis

Clinical phenotype and expression data for unmatched healthy and tumour samples were extracted from the TCGA and GTEx data, available in the TCGA-TARGET-GTEX cohort from the University of California Santa Cruz public repository, Xena [22]. The data were processed by UCSC from raw RNAseq reads using the TOIL pipeline ensuring the quantification and cross-dataset normalisation of gene expression without any computational batch effects [23]. All gene expression values are presented in units of log_2_(norm_count + 1). Differential expression between normal and tumour sites was calculated as fold change, defined as the ratio between tumour and healthy tissue samples. In this study we used only primary tumour data, and further filtered the samples to remove those lacking expression information for the genes of interest. The summary of analysed samples is shown in Table 1. While we have used the largest uniformly processed and normalized dataset available for tumour and healthy tissue samples, we are aware that using a single data set can influence the outcome of our results and that this is a limitation of our study. However, we addressed this issue by evaluating the potential for analysis artifacts by splitting the data into two randomly selected groups and measuring the change in the average expression levels for our genes of interests (Appendix A). Our results show no change between the two groups suggesting that there is little chance for a data bias to impact the biological insights we have drawn from the data.

### 2.2. Prediction of Targets

The target prediction for *MNX1*-*AS1* and *MNX1*-*AS2* was performed on RNAInter 4.0 [24] by selecting the top 200 gene from the RNA-Protein hits based on the score. The top 200 targets of *MNX1* were extracted from JASPAR database of transcription factor binding sites [25] based on the score. The Venn diagrams were generated on VennPainter [26].

### 2.3. Differential Expression Analysis

RNA-sequencing values of transcripts were downloaded in counts units from the University of California Santa Cruz public repository [22]. The differential expression analysis was performed using limma (*limma* package version 3.48.3 in R statistical software environment version 4.0.3-The R Foundation for statistical Computing, Vienna, Austria implemented in R studio version 1.4.1717, RStudio, Boston, MA, USA). The statistically significant genes were extracted by filtering by *p* adjusted value < 0.01.

### 2.4. Gene Ontology Analysis

The Gene Ontology (GO) enrichment analysis and network construction was performed on ExpressAnalyst (available at www.expressanalyst.ca, last accessed on 20 October 2022) using the PANTHER Biological Process database. The significance was determined using *p* value < 0.05.

### 2.5. Statistical Analysis

All statistical analyses were carried out using the standard libraries (*stats* 4.0.2, *ggpubr* 0.4.0) in R (statistical software environment version 4.0.3-The R Foundation for statistical Computing, Vienna, Austria implemented in R studio version 1.4.1717, RStudio, Boston, MA, USA). The Wilcoxon test was used to evaluate the statistical significance, with a set threshold for significance set to a *p* value ≤ 0.05. The correlation between clinicopathological features and expression levels was calculated using the Chi-squared test. The gene expression was defined as “high” or “low” based on the absolute expression value relative to the mean average of the respective phenotype. The receiver-operating characteristic (ROC) was used to calculate the area under the curve (AUC) value to evaluate the diagnostic power in differentiating the tumour vs. normal samples.

### 2.6. Clustering Analysis

We used T-distributed Stochastic Neighbour Embedding (t-SNE), as implemented using the library *Rtsne* 0.15, to visualise and identify the differentiation between various phenotypes based on expression data. The t-SNE parameters were adjusted for perplexity (5–50) and numbers of iterations (5000) according to sample sizes, abiding to the ranges suggested by van der Maaten et al. [27]. The principal component analysis (PCA) was implemented in the R (statistical software environment version 4.0.3-The R Foundation for statistical Computing, Vienna, Austria implemented in R studio version 1.4.1717, RStudio, Boston, MA, USA) environment using the *factoextra* package version 1.0.7 for the analysis and identification of gene clusters. The contribution for each principal component (PC) was considered significant above the threshold of 33%.

### 2.7. Survival Analysis

The clinical outcomes were extracted from survival data for the TCGA cohort available for 10,496 samples [22]. The following survival parameters were included: overall survival (OS), disease-specific survival (DSS), disease-free interval (DFI), and progression-free interval (PFI). The survival probability was analysed using Kaplan–Meier plots generated using R (statistical software environment version 4.0.3-The R Foundation for statistical Computing, Vienna, Austria implemented in R studio version 1.4.1717, RStudio, Boston, MA, USA) libraries *survminer* (0.4.9) and *survival* (3.2-11). The expression data were segregated in “high” and “low” by comparison to the average expression recorded by the mean in each analysed group.

## 3. Results

### 3.1. MNX1, MNX1-AS1, and MNX1-AS2 Are Dysregulated in Most Cancers

We explored the expression profile of *MNX1*, *MNX1*-*AS1*, and *MNX1*-*AS2* in 18 cancer tissues from the TCGA dataset (Table 1) and unmatched healthy samples from GTEx (Figure 1B).

The majority of cancers analysed showed a statistically significant increase in expression for all three genes in tumour samples. The highest fold changes were recorded for *MNX1* and *MNX1*-*AS1* in UCEC, PRAD, LUAD, ESCA, and COAD/READ (Figure 1C). By contrast, *MNX1*-*AS2* showed a modest upregulation in expression for the majority of cancers. Moreover, cancers of the gastrointestinal (GI) tract showed a rather homogenous expression profile across the three genes. LIHC, STAD, TGCT, and THCA were the only tumours that showed different expression profiles across the three genes in terms of both statistical significance as well as the differential expression (up vs. down regulation). *MNX1* expression was unchanged between normal and tumour in LIHC and THCA, however *MNX1*-*AS1* and *MNX1*-*AS2* showed a statistically significant upregulation. Notably, all three genes were significantly downregulated in pancreatic adenocarcinoma.

The average expression values in the different cancers revealed a similarity in expression patterns by hierarchical clustering (Figure 1D). READ, COAD, STAD, PAAD, and TGCT showed the most similar expression pattern characterised by a high expression of all three genes, with the highest expression being *MNX1*. A second cluster of tumours showed similar expression levels between *MNX1* and *MNX1*-*AS1* and included PRAD, LUAD, ESCA, UCEC, BRCA, and LUSC. Low expression values of all three genes clustered SKCM, UCS, GBM, LGG, OV, LIHC, and THCA. Furthermore, we have explored the correlate of the expression profiles of the any of the three genes in the 18 studied cancers (Appendix A). The results show a cancer/tissue specific relationship in particular for the associations with *MNX1*-*AS2*.

### 3.2. Diagnostic Biomarker Potential for MNX1, MNX1-AS1, and MNX1-AS2

Given the differential expression profile shown by the three genes across a large variety of cancer types, we investigated their potential to be defined as individual or collective diagnostic biomarkers for cancer. We performed a ROC analysis to evaluate the sensitivity and specificity of the three genes as a collective diagnostic biomarker (Figure 2). The combinatorial expression profile of *MNX1* and its associated lncRNAs (“combo”) showed high AUC values for most cancers analysed, indicating a high sensitivity and specificity as a collective diagnostic marker. Cancers of the GI system achieved the best AUC scores, followed closely by LUAD, TGCT, PRAD, and UCEC. By contrast, none of the individual genes, nor their combined expression profiles produced an AUC higher than 0.7 in SKCM, THCA, GBM, and LIHC.

Overall, *MNX1*-*AS2* showed the lowest performance of AUC values across all cancer sites, with the exception of PAAD, where *MNX1*-*AS2* was comparable and even exceeded AUC values of *MNX1* and *MNX1*-*AS1*. High level AUC scores were observed for *MNX1*-*AS2* in PRAD and UCEC suggesting a moderate diagnostic power for these tumour types.

Interestingly, in several sites (e.g., COAD, READ, LUAD, and LUSC) the lncRNAs showed a superior performance to *MNX1*, suggesting a higher tissue specificity of the antisense transcripts in these tumours. Despite lower individual values, the combination expression patterns displayed the highest performance in the majority of cancers, indicating that the transcriptional patterns of *MNX1*, *MNX1*-*AS1*, and *MNX1*-*AS2* are important determinants in distinguishing normal vs. tumour samples.

We used t-SNE analysis to assess whether the combinatory expression patterns of *MNX1*, *MNX1*-*AS1*, and *MNX1*-*AS2* have discriminatory power in distinguishing tumour vs. normal tissue samples (Figure 3A). Overall, we found that the three genes are able to differentiate between the two states in a tissue specific manner (Figure 3A). Specifically, patients with GI tract tumours were clearly distinguished from health individuals using the combined expression profiles of the *MNX1* and its associated antisense transcripts. A poorer discriminatory power was observed for the other tumour types, namely TGCT, LUAD, LUSC, UCS, and UCEC. By contrast, BRCA, THCA, OV, LGG, GBM, SKCM, and LIHC cancers show no differentiation from the corresponding normal samples based on the transcription profiles of the three genes, reflecting the low expression and fold change at these sites.

We then quantified the observed t-SNE clustering and differentiation of tumour and normal samples using PCA. The contribution percentage of each gene for each principal component (PC) was extracted and is shown in Figure 3B. In the majority of tumours, *MNX1* and *MNX1*-*AS1* are the main contributors (i.e., accounting for more than 33% of the total contribution) to PC1, while *MNX1*-*AS2* is the highest contributor for PC2. PAAD and TGCT diverged from this trend, with *MNX1* being the sole significant contributor to PC1 and both *MNX1*-*AS1* and *MNX1*-*AS2* contributing to PC2. Consistent with the fold change patterns, PCA contribution reflected the changes in expression and the tissue specificity of the lncRNAs.

### 3.3. MNX1, MNX1-AS1, and MNX1-AS2 Correlate with Oncologic Clinicopathological Features

Next, we looked at the association between clinicopathological features and the expression levels of *MNX1*, *MNX1*-*AS1*, and *MNX1*-*AS2* (summarized in Figure 4). Statistically significant associations are reported in detail in Appendix A. The strongest associations in terms of X^2^ and *p* values were found for disease subtype in lung tumours (LUAD and LUSC), ESCA, UCEC, and TGCT. High expressions of all three genes were able to discern adenocarcinomas from squamous tumours in LUAD vs. LUSC, as well as in ESCA. A weaker association with adenocarcinomas was also identified in COAD, however for high *MNX1* and *MNX1*-*AS1* expressions only. In UCEC, high expressions of the three genes were also associated with adenocarcinomas, while lower expressions defined cystic, mucinous, and serous types. High expressions of *MNX1*, *MNX1*-*AS1*, and *MNX1*-*AS2* also correlated with the TGCT subtype of seminomas. The high expression of *MNX1*-*AS1* was only associated with oligodendrogliomas in LGG.

We also observed a correlation between increased expression levels and clinical features associated with advanced disease progression (i.e., TNM features, staging, and grading), which was particularly evident for GI tract cancers. High expressions of all three genes correlated with a higher histological grade for ESCA and the presence of lymph node invasion in both ESCA, PRAD, and LIHC. In PRAD, high expression of all three genes also correlated with larger tumours. On the contrary, high expression of *MNX1*, but not of the antisenses, was associated with a smaller tumour size in PAAD.

Other clinical features mainly associated with *MNX1* and *MNX1*-*AS1* expressions. In COAD, *MNX1*, and *MNX1*-*AS1* high expression correlated with the presence of metastasis. Lymph node involvement was associated with increased expressions of these two genes in LUSC and LIHC. *MNX1* and *MNX1*-*AS1* also showed an association with age, with higher expression values in patients above the age of 50 years in BRCA, LGG, LIHC, and PRAD. Late disease staging was associated with increased expressions of *MNX1* and *MNX1*-*AS2* in COAD and a lower expression at early stages of LIHC.

Overall, *MNX1* expression had the highest number of statistically significant associations (*n* = 25), followed by *MNX1*-*AS1* (*n* = 22) and *MNX1*-*AS2* (*n* = 14). No clinicopathological features were significantly linked to any of the *MNX1* transcripts in UCS, SKCM, THCA, and READ.

### 3.4. MNX1, MNX1-AS1, and MNX1-AS2 Distinguish Tumour Subtypes

Building on the positive results showcased by the discriminatory power of the three genes to distinguish between tumour and healthy samples, we expanded our analysis to evaluate the biomarker potential in differentiating various histological subtypes. The observation of defined tumour clusters in ESCA, STAD, TGCT, LUAD, and LUSC using t-SNE (Figure 3) suggested that specific tumour subtypes may be recognised by the combined expression profiles of the *MNX1*, *MNX1*-*AS1*, and *MNX1*-*AS2*. The Chi-square analysis of clinicopathological features (Figure 4) also revealed a defining power of these genes in distinguishing disease subtypes.

First, we combined multiple anatomical sites with adenocarcinomas, squamous tumours, seminomas, and non-seminomas. *MNX1*, *MNX1*-*AS1*, and *MNX1*-*AS2* were capable of discerning adenocarcinomas and squamous carcinoma samples by combinatorial analysis visualised using t-SNE. Testicular seminomas and non-seminomas also appeared distinctly clustered (Figure 5A). When colour-coded by anatomical site, adenocarcinomas of the GI tract (COAD, ESCA, PAAD, STAD, and READ) clustered closely together and were discernible from squamous ESCA tumours, LIHC, LUAD, and LUSC. TGCT seminomas formed a defined cluster within the majority of GI adenocarcinomas (Figure 5B).

This separation was also confirmed by hierarchical clustering on individual samples, classified by anatomical site and tumour type (Figure 5C and Appendix A). The vast majority of adenocarcinomas of the GI organs were characterised by a high expression of the three genes. A second major cluster grouped squamous ESCA tumours, both LUAD and LUSC, and LIHC. The distinct seminoma TCGT clustering is also evident from the central cluster of the heatmap, which separates seminomas from non-seminomas. These results indicate that *MNX1*, *MNX1*-*AS1*, and *MNX1*-*AS2* are expressed at a higher level in adenocarcinomas and seminomas, compared to squamous and non-seminoma tumours, and that their discriminative power is visualisable using clustering methods.

Site-specific clustering using t-SNE revealed specific tumour clusters in TGCT and ESCA (Figure 5E–H), but weakly in COAD, LUAD/LUSC, and UCEC (Appendix A). A summary of the results from Chi-square and t-SNE analyses is reported Figure 6D. *MNX1*, *MNX1*-*AS1*, and *MNX1*-*AS2* could distinguish seminomas against embryonal carcinomas in TGCT (Figure 5E) and adenocarcinomas against squamous subtypes in ESCA (Figure 5G–H). Detailed analysis of histological subtypes of TGCT revealed that the TGCT of the mixed subtype were equally distributed between the seminoma and non-seminoma clusters (Figure 5E), which is coherent with the biological properties of this subtype, yet clearly distinguishable from the corresponding healthy testicular tissue. ESCA tumours of the squamous type, on the other hand, showed overlap with healthy tissue of the oesophagus (Figure 5G), but a remarkable distinction when analysed in tumour tissues only (Figure 5H).

### 3.5. Prediction of Biological Functions of MNX1, MNX1-AS1, and MNX1-AS2 in Cancer

We attempted to explore biological functions of the two antisense transcripts by looking at the potential targets of *MNX1*-*AS1* and *MNX1*-*AS2*. We observed a large degree of overlap between the predicted targets of *MNX1*-*AS1* and *MNX1*-*AS2* and a smaller set of antisense specific gene targets (Figure 6A). As limited functional studies are available for these transcripts, we inferred their association with biological processes using Gene Ontology (GO) analysis (Figure 6B–D). Both *MNX1* and antisenses shared several biological processes related to transcription and developmental processes (e.g., anatomical structure morphogenesis, endoderm development, nervous system development), which supports a role in gene expression regulation during development. Notably, targets of *MNX1*-*AS1* were enriched in processes related to proliferation, cell cycle regulation, apoptosis, and angiogenesis (Figure 6C). *MNX1*-*AS2* also targets genes associated with the cell cycle and angiogenesis, as well as a unique association with epigenetic regulation of gene expression and chromatin remodelling (Figure 6D).

Based on the high scores observed in the chi-square analysis of clinicopathological features (Figure 4) and subtype prediction using t-SNE (Figure 5), we performed differential expression analysis of COAD, ESCA, PRAD, TGCT, LUSC, and UCEC to identify biologically relevant differences between samples expressing high and low *MNX1*, *MNX1*-*AS1*, and *MNX1*-*AS2*. In agreement with some redundancy of predicted functions (Figure 6E), we observed overlap between processes for *MNX1* and the antisenses, but also unique terms associated with specific antisenses in a cancer-specific fashion. The most reoccurring GO terms for all three genes encompassed biological processes relating to cellular transport, metabolism, transcription, and cellular differentiation. The patterns of biological process enrichment indicate varying roles for the two antisenses, which are likely modulators of transcription to accomplish changes in cellular metabolism and identity. In agreement with the main clinical feature associated with high expressions of *MNX1* and antisenses being disease subtype, we observed cell differentiation and lipid metabolism particularly enriched in UCEC and ESCA. We also observed the enrichment of nervous system processes (nervous system development and synaptic vesicle endocytosis), which involve the expression of cell adhesion genes that can modulate invasion behaviour. In TGCT, the highest enriched process was spermatogenesis, which suggests a modulation of testicular gene expression patterns that distinguishes seminomas from non-seminoma tumours.

### 3.6. Prognostic Biomarker Potential for MNX1, MNX1-AS1, and MNX1-AS2

In order to uncover potential prognostic value of *MNX1*, *MNX1*-*AS1* and *MNX1*-*AS2*, we performed Kaplan–Meier analysis based on their expression levels in all cancers. The individual expression levels of *MNX1*, *MNX1*-*AS1*, and *MNX1*-*AS2* had a weak correlation with survival (overall—OS, disease free—DFI, and progression free—PFI) (Appendix A). However, statistical significance for segregating the patient prognostic profiles using gene expression was reached in colon, stomach, ovarian, lung (LUSC, LUAD) and brain (LGG) cancers (Figure 7). Higher expressions of all three genes were associated with inferior OS in COAD. A higher *MNX1*-*AS2* expression was associated with PFI in COAD (*p* = 0.033) and LGG (0.028). Interestingly, in stomach cancer, a lower expression of *MNX1* associated with inferior clinical outcomes for both OS and PFI. A similar trend was observed for low expressions of *MNX1*-*AS1* in OV for poor outcomes in terms of OS and DFI. In lung cancers LUAD and LUSC, only a high *MNX1* expression was correlated with inferior DFI and PFI. Overall, these observations indicated that individual *MNX1*, *MNX1*-*AS1*, and *MNX1*-*AS2* expressions harbour modest prognostic potential in selected tumours. Combinatorial analysis of expressions, however, showed stronger statistical differences in OS for COAD (*p* = 0.018), ESCA (*p* < 0.0001), and LGG (*p* < 0.001), suggesting a value in multi-gene expression combinations (Figure 8). In particular, in all three cancers, the combinatorial high expression of *MNX1*, *MNX1-AS1*, and *MNX1*-*AS2* indicated poor clinical outcomes.

## 4. Discussion

Several reports have highlighted *MNX1* and *MNX1*-*AS1* dysregulation in various malignancies [9,10,11,12,13,14,15,16,17,18,19,20,21], however the combined expression patterns of the locus comprising the *MNX1* gene and its associated lncRNAs *MNX1*-*AS1* and *MNX1*-*AS2* have not been explored. Here, we leveraged the TCGA dataset to show the clinical potential of the three genes in cancer, in order to address the clinical need for more accurate diagnostic and prognostic biomarkers.

We showed that *MNX1*, *MNX1*-*AS1*, and *MNX1*-*AS2* expressions are dysregulated in most cancers analysed, with particularly high fold changes observed in uterine, prostate, oesophageal, colorectal, and lung cancers. The significant differential expression between normal and tumour tissues suggested that these genes can serve as hallmark diagnostic markers for various cancers. These observations are in line with previous reports describing overexpression of *MNX1* in cancers of the colon [15], breast [11,12], prostate [13], and glioma [17], as well as *MNX1*-*AS1* in breast [28], colon [29], oesophageal [30], stomach [31], glioblastoma [32], liver [33], lung [34], ovarian [35,36], and prostate [37] tumours. *MNX1* was not found to be significantly overexpressed in liver cancer samples from TCGA compared to healthy liver tissue from GTEx, despite reports of upregulation in hepatocellular carcinoma [18]. Nevertheless, *MNX1* and its antisenses showed statistically significant associations in grading, lymph node invasion, and staging in liver cancer. One of the most interesting results was the overexpression of *MNX1*, *MNX1-AS1*, and *MNX1*-*AS2* in testicular tumours and their downregulation in pancreatic adenocarcinomas, which have not been previously reported.

In particular, we describe here for the first time the dysregulation of *MNX1*-*AS2* in cancer. While its fold changes between normal and tumour samples were lower than *MNX1* and *MNX1*-*AS1*, its expression levels were particularly high in gastro-intestinal tract and testicular tumours, with the highest negative fold change in pancreatic adenocarcinoma. The current literature on *MNX1*-*AS2* is scarce, with only one report on *MNX1*-*AS2* dysregulation associated with pre-eclampsia [38]. From our study, we identified individual and combinatorial diagnostic and prognostic potential of *MNX1*-*AS2* in several cancers. From the PCA analysis, *MNX1*-*AS2* showed a significant contribution in sample segregation in both pancreatic and testicular tumours. In terms of association with clinical features, *MNX1*-*AS2* expression shows a strong discrimination of disease subtype (adenocarcinomas vs. squamous, or seminoma vs. non-seminoma).

Overall, *MNX1*, *MNX1-AS1*, and *MNX1*-*AS2* individual expressions exhibited limited prognostic potential, compared to their more widespread diagnostic power. Poor clinical outcomes have been previously associated with the overexpression of *MNX1*-*AS1* in various cancers [21]. In this study, while some association was found with OS, DFI, and PFI in colon, stomach, ovarian, lung, and brain cancers, stronger statistical differences were observed in combinatorial analyses of their expressions in colon and oesophageal cancers and lower grade glioma. The disparity in the distribution of clinical stages in the analysed cohorts may have impacted the evaluation of prognostic significance and especially in tumours where the range of overexpression of the genes is homogeneous (e.g., COAD).

Thanks to the plethora of transcriptomics data becoming available, lncRNAs are increasingly gaining clinical interest for their expression dysregulation and involvement in biological processes related to cancer. Nonetheless, the functions and mechanisms of the vast majority of identified lncRNAs remain to be elucidated [39,40]. The high tissue-specificity in expression renders lncRNAs attractive markers for specific cellular states, with potential correlations with specific disease characteristics [41,42]. The tissue tropism of lncRNA expressions is in fact an advantage in providing biomarker specificity. As revealed by the ROC analysis, the expression of the associated lncRNA *MNX1*-*AS1* and *MNX1*-*AS2* performed better than the corresponding coding gene in distinguishing cancerous vs. normal samples in some tissues. In virtue of lncRNA’s regulatory function in gene expression modulation, the transcriptional relationship between coding genes and non-coding transcripts may also reveal biologically relevant clues on the transcriptional activity of specific loci [43,44]. In fact, in this study, combinatorial expression patterns displayed a high degree of accuracy in distinguishing tumour samples against normal tissues and tumour subtypes. Nevertheless, the combinatorial predictive power may only be clinically applicable in specific tumours rather than a collective pan-cancer biomarker.

While the functions of *MNX1*-*AS1* and *MNX1*-*AS2* remain elusive, we identified a high degree of overlap in predicted targeted genes and biological functions associated with the two antisense. By comparing the DEGs in high vs. low expressing patient samples, we observed different target genes but similar biological processes across tumours, mainly associated with cellular differentiation and metabolism. We can speculate that these genes function as selective transcriptional regulators with overlapping targets, with tissue/cancer-specific action. From the observations of the transcriptional elements regulating the expression of these genes (Figure 1A), distinct promoters and enhancer regions allow the selective expression of *MNX1* or its transcripts to modulate the regulation of their targets. In cancer, the dysregulation of this physiological transcriptional balance results in the activation or repression of key cellular processes.

The implementation of dimensionality reduction techniques (e.g. clustering methods such as t-SNE [27]) allowed us to uncover gene relationships with ease of visualisation and interpretation. Specifically, we were able to capture tissue-specific transcriptional patterns of the *MNX1* locus that are altered in the cancer state, with a higher specificity than measuring the expression levels of the individual genes. Interestingly, the *MNX1*, *MNX1-AS1*, and *MNX1*-*AS2* combined expression patterns defined specific tumour subtypes, displaying both tissue- and cell-type specificity of adenocarcinomas and seminomas. Transcriptional and molecular differences between adenocarcinomas and squamous cancers have been identified, with a high degree of similarity even across different anatomical sites [45]. While the role of *MNX1*-associated lncRNAs has not been investigated during embryonic development, *MNX1*’s role in instructing pancreatic beta cell identity (endocrine portion of the pancreas) may explain their reminiscent expression in the healthy tissues of the gastro-intestinal tract and the specificity in expression in glandular tumours. In addition to providing diagnostic specificity for these cancers, the dysregulation of the *MNX1* transcripts may also be interesting in the light of tumour genesis and development.

While the association with GI organs may derive from the embryonic functions of *MNX1*, the expression of *MNX1* and its antisenses in normal testis and testicular cancer does not seem to link with developmental activity. *MNX1* has not been described in genital development but has been reported to be expressed in the developing urinary tract [46]. It must be noted that testes are known to be a transcriptionally active tissue, with a large proportion of genes being expressed [47,48]. Nevertheless, *MNX1*, *MNX1-AS1*, and *MNX1*-*AS2* transcription profiles exhibited high specificity in differentiating seminomas from non-seminomas samples. The expression patterns of *MNX1* and its lncRNA’s in these subtypes may be linked to the fact that the cell of origin of seminomas and non-seminomas appears to be distinct, with unique transcriptional identities [49]. Moreover, the distinction between the two subtypes also has therapeutic implications due to the radiosensitivity of seminomas compared to non-seminomas [50]. Thus, the use of the three genes as collective biomarkers with subtype discriminatory power can strongly impact the patient’s course of treatment. Thus, it would be an interesting avenue to pursue in the future examination of the transcriptional activity based on the cellular subtype in the tumour environment.

The use of a next-generation sequencing (NGS) platform in clinical practice will enable these observations to be translated into diagnostic procedures [51]. Overall, *MNX1*, *MNX1*-*AS1*, and *MNX1*-*AS2* show a robust performance in distinguishing normal samples from tumours, with a good site-specific diagnostic power. The use of a simple three gene signature has the potential to improve histological classification workflows and showcase lncRNAs as diagnostic markers [52,53,54]. Currently, the use of NGS data has already shown promising results in supporting histopathological information [51,55,56] and the clinical potential of *MNX1* and its antisenses.

## 5. Conclusions

The availability of transcriptomics data and the use of an NGS platform in clinical applications will enable the use of both coding and non-coding transcripts as clinical individual and collective biomarkers. In this study, we identified the expression patterns of the *MNX1* gene and its antisense transcripts *MNX1*-*AS1* and *MNX1*-*AS2* as potential diagnostic and prognostic biomarkers in cancer. Based on the analysis of the TCGA cohort, these genes are dysregulated in the majority of cancers. By the use of clinicopathological features, clustering, and ROC analysis, we identified a strong diagnostic power for both individual and combinatorial expressions of these three genes in distinguishing tumour vs. healthy samples, as well as differentiating histological tumour subtypes. In particular, *MNX1*, *MNX1-AS1*, and *MNX1*-*AS2* could discern adenocarcinomas from squamous tumours at various sites, as well as testicular seminomas from non-seminomas. As the use of next generation sequencing platforms widens in the clinic, we foresee that robustly differentially expressed genes with tissue specificity, such as *MNX1* and its antisense transcripts, will be ideal diagnostic markers to be implemented in current clinical workflows.

## Figures and Tables

**Figure 1 cells-11-03577-f001:**
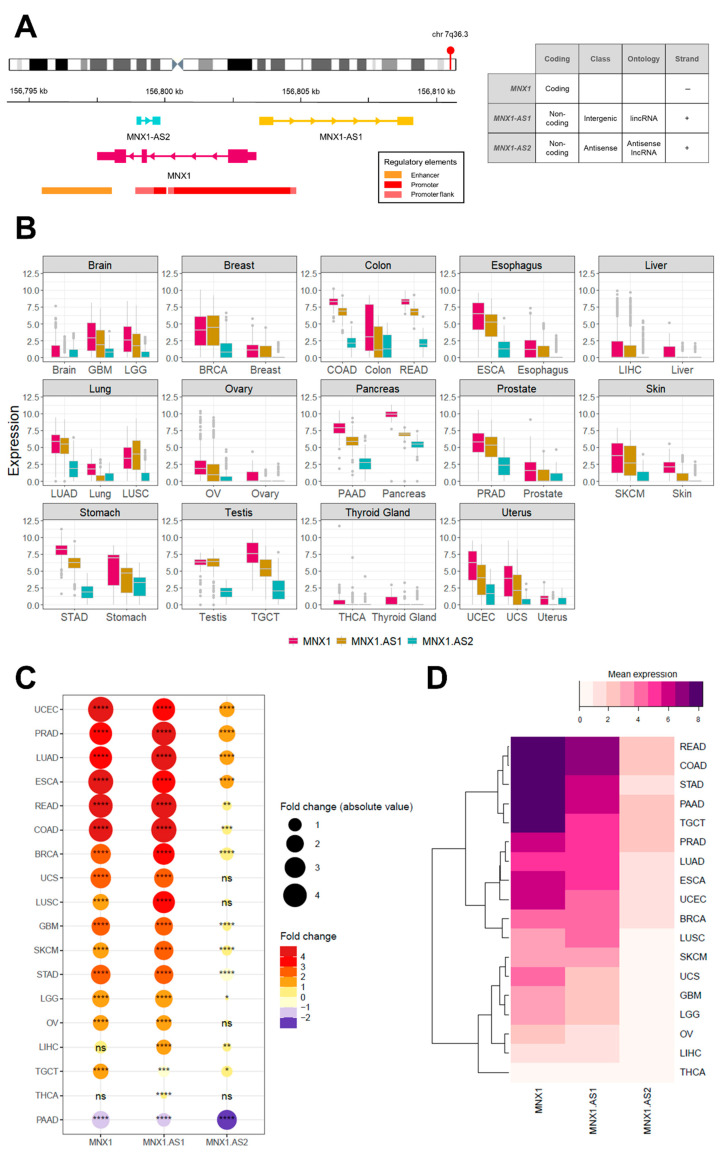
Overview of expression of *MNX1*, *MNX1*-AS1, and *MNX1*-*AS2* in cancers of the TCGA cohort. (**A**) Schematic representation of the chromosome 7q36.3 locus and location of the *MNX1* gene and its associated non-coding transcripts *MNX1*-*AS1* and *MNX1*-*AS2*. Arrows within the gene structure indicate the direction of transcription and vertical boxes represent gene exons. Underneath, known regulatory elements are shown. The table on the right summarises the classification and genomic features of the three genes. (**B**) Expression values of *MNX1*, *MNX1*-*AS1*, and *MNX1*-*AS2* by anatomical site in cancer (indicated by TCGA acronyms) and normal tissues (indicated in lower case). (**C**) Fold change in expression between tumour (TCGA) and normal (GTEx) by tumour site and statistical significance of the differential expression. The size of the dots represents the absolute fold change value and the colour gradients indicate increased (positive) or decreased (negative) fold change. (**D**) Hierarchical clustering of TCGA cancer by mean expression values of *MNX1*, *MNX1*-*AS1*, and *MNX1*-*AS2* revealed similarities in patterns of expression of the three genes. +: positive strand; –: negative strand; Statistical significance is symbolised as: *p* value < 0.05 (*), 0.001 (**), 0.0001 (***), 0.00001 (****), ns: not significant.

**Figure 2 cells-11-03577-f002:**
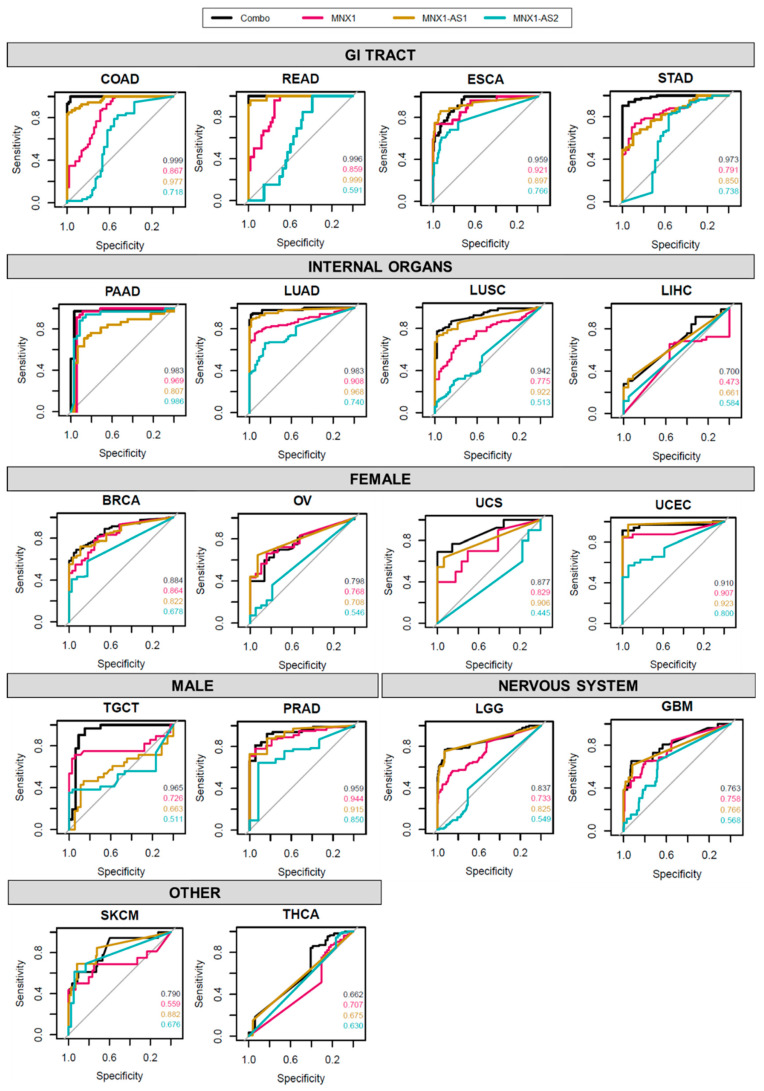
ROC analysis of *MNX1*, *MNX1*-*AS1*, and *MNX1*-*AS2* expressions as diagnostic evaluators. Each plot represents the ratio of sensitivity and specificity of the discriminatory power for each gene and their combination. The AUC values for each gene and combination are shown on the bottom right corner, with values closer to 1 indicating a higher performance in discerning normal vs. tumour samples.

**Figure 3 cells-11-03577-f003:**
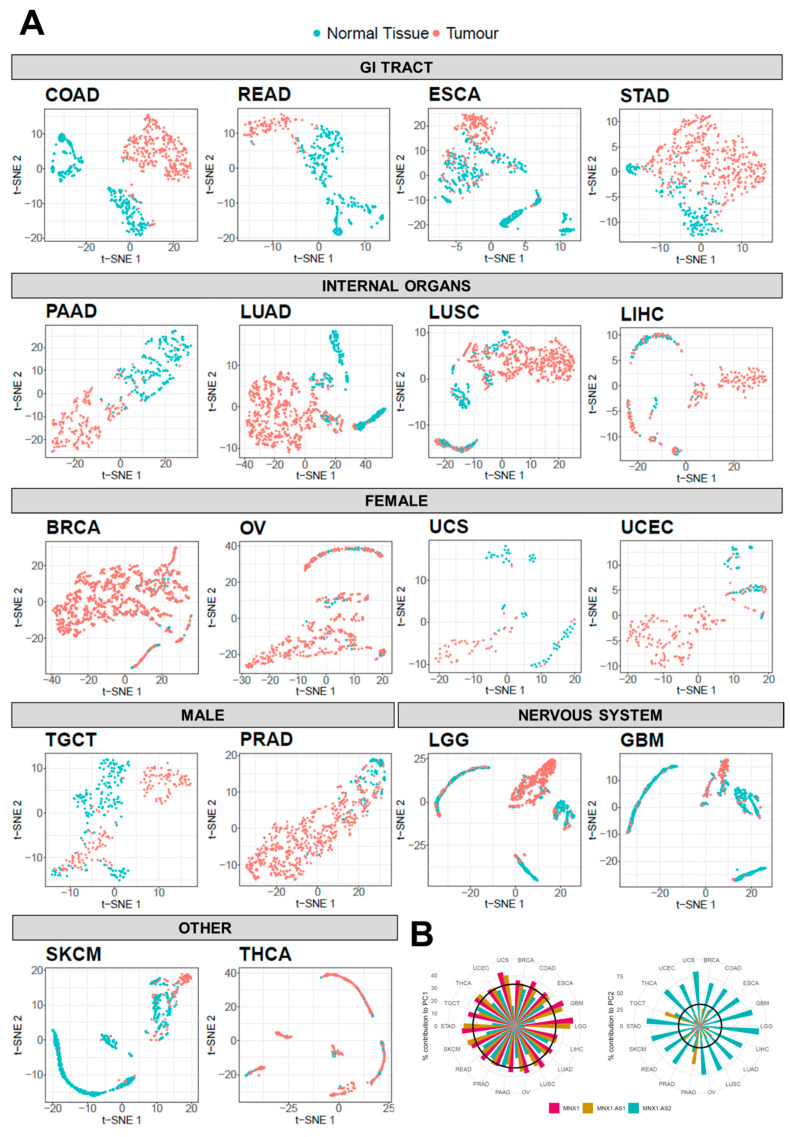
t-SNE clustering analysis of normal vs. tumour samples based on the combinatorial expressions of *MNX1*, *MNX1*-*AS1*, and *MNX1*-*AS2*. (**A**) The visualization of multi-gene relationships using t-SNE plots allowed the discrimination of tumour vs. normal samples in several cancer sites. A distinct clustering of normal (in green) and tumour (in red) samples, with no or minimal overlap between samples, defined a strong performance in diagnostic power. (**B**) PCA contribution of expressions of *MNX1*, *MNX1*-*AS1*, and *MNX1*-*AS2* in clustering differentiation of cancer and normal samples. Percentage of contribution to PC1 (left) and PC2 (right). The black intercept indicates the significance threshold of 33%.

**Figure 4 cells-11-03577-f004:**
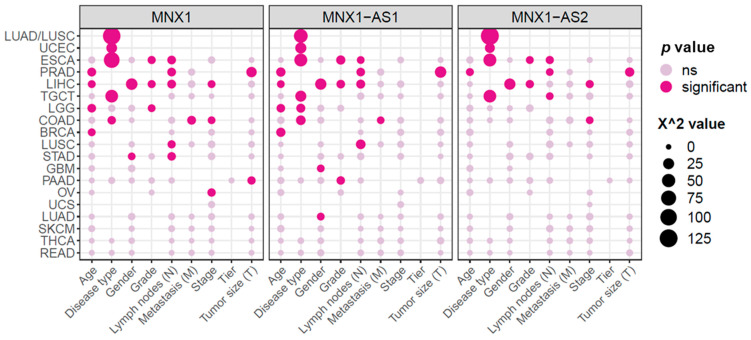
Summary of correlations between *MNX1*, *MNX1*-*AS1*, and *MNX1*-*AS2* expressions and clinicopathological features. Each dot represents the value of Chi-square correlation expressed in X^2^ value, as well as the statistical significance in pink (at a threshold of *p* value < 0.05). Detailed sample information on statistically significant features is reported in Appendix A. ns: not significant.

**Figure 5 cells-11-03577-f005:**
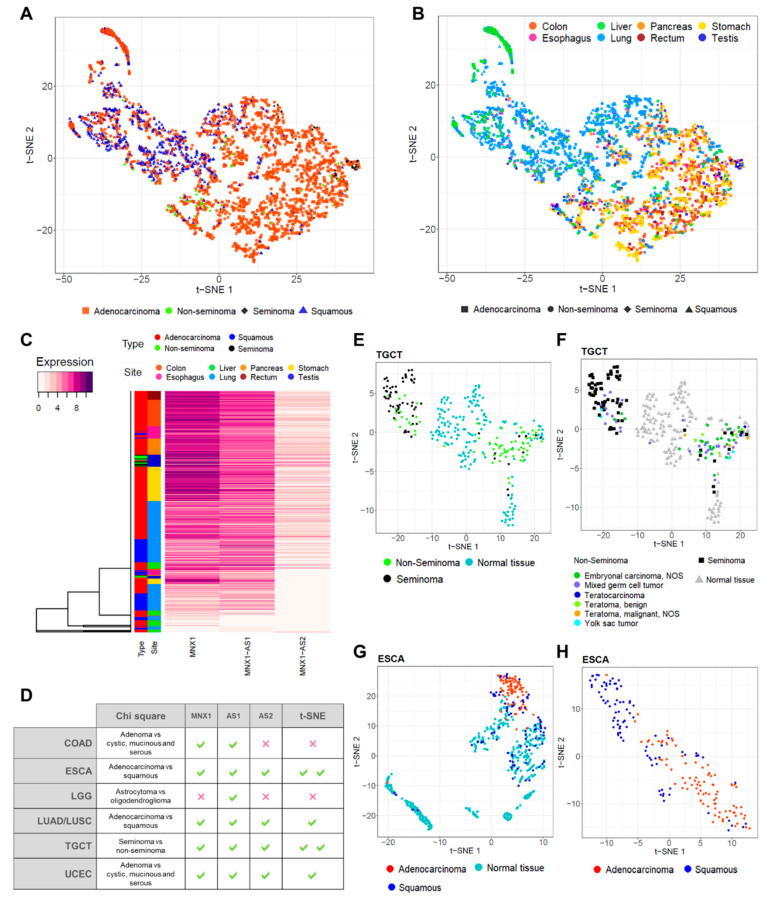
Site-specific t-SNE clustering for the identification of disease subtypes by combinatorial expressions of *MNX1*, *MNX1*-*AS1*, and *MNX1*-*AS2*. (**A**) t-SNE visualisation of selected cancer sites by tumour type (adenocarcinoma, squamous, non-seminoma, and seminoma). (**B**) t-SNE clustering by tumour type and colour-coded by cancer site. (**C**) Hierarchical clustering of *MNX1*, *MNX1*-*AS1*, and *MNX1*-*AS2* expression levels, with cancer type and site information, revealing similar clustering patterns to t-SNE analyses. (**D**) Comparison between Chi-square results of distinction between disease subtypes and the correspondence in t-SNE visualisation of distinct clusters. One green tick indicates a statistically significant correlation in Chi-square tests and the ability to differentiate disease subtypes using t-SNE. Two green ticks indicate a strong t-SNE clustering performance. (**E**) Site-specific t-SNE clustering of TGCT tumours against healthy testicular tissue (GTEx) by TGCT types of seminomas and non-seminomas. (**F**) Diagnostic subtype specification of samples within clusters identified using t-SNE in TGCT seminomas and non-seminomas. (**G**) t-SNE clustering of ESCA tumour subtypes (adenocarcinomas and squamous carcinomas) and healthy oesophageal tissue. (**H**) t-SNE clustering of ESCA adenocarcinomas and squamous carcinomas by analysis of tumour tissues only.

**Figure 6 cells-11-03577-f006:**
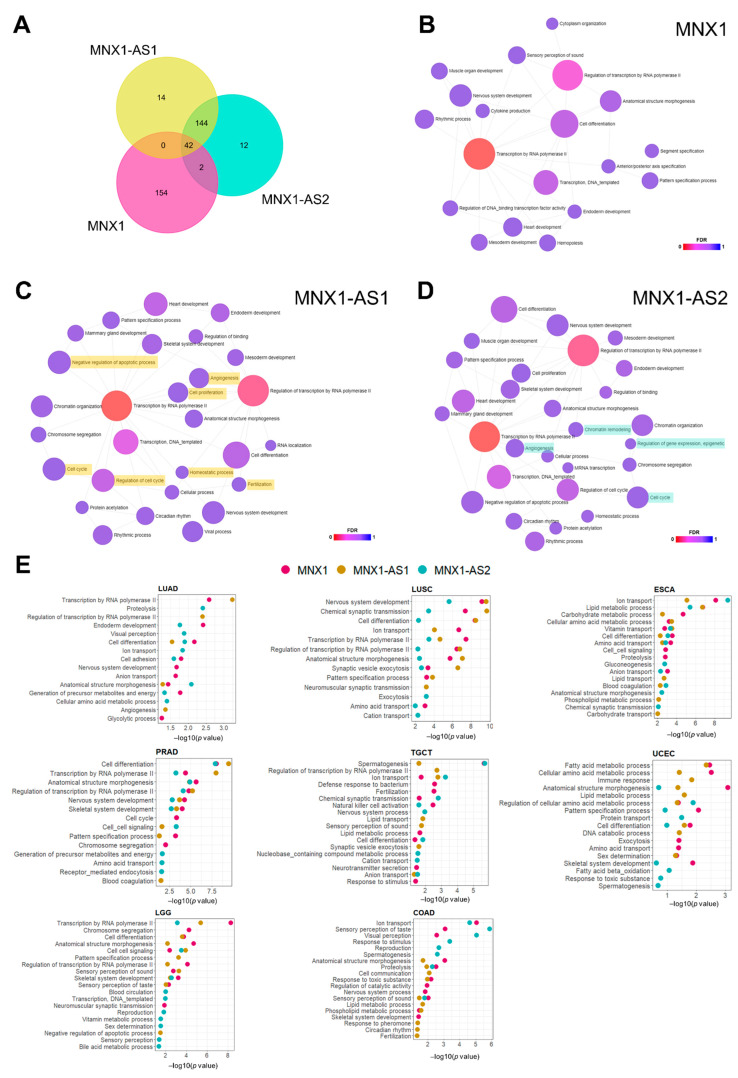
Functional prediction of *MNX1*, *MNX1*-*AS1*, and *MNX1*-*AS2* functions in cancer using Gene Ontology (GO) analysis. (**A**) Venn diagram of shared and unique predicted targets of *MNX1*, *MNX1*-*AS1*, and *MNX1*-*AS2*. (**B–D**) Biological processes associated with targets of *MNX1* (**B**), *MNX1*-*AS1* (**C**), and *MNX1*-*AS2* (**D**), represented as network of biological processes terms from the PANTHER database and coloured by false discovery rate (FDR; shades of red representing lower values). Highlighted processes are unique to *MNX1*-*AS1* (yellow) or *MNX1*-*AS2* (turquoise). (**E**) Top 10 biological processes (PANTHER) enriched in differentially expressed genes of tumours expressing high levels of *MNX1* (in pink), *MNX1*-*AS1* (in yellow), and *MNX1*-*AS2* (turquoise).

**Figure 7 cells-11-03577-f007:**
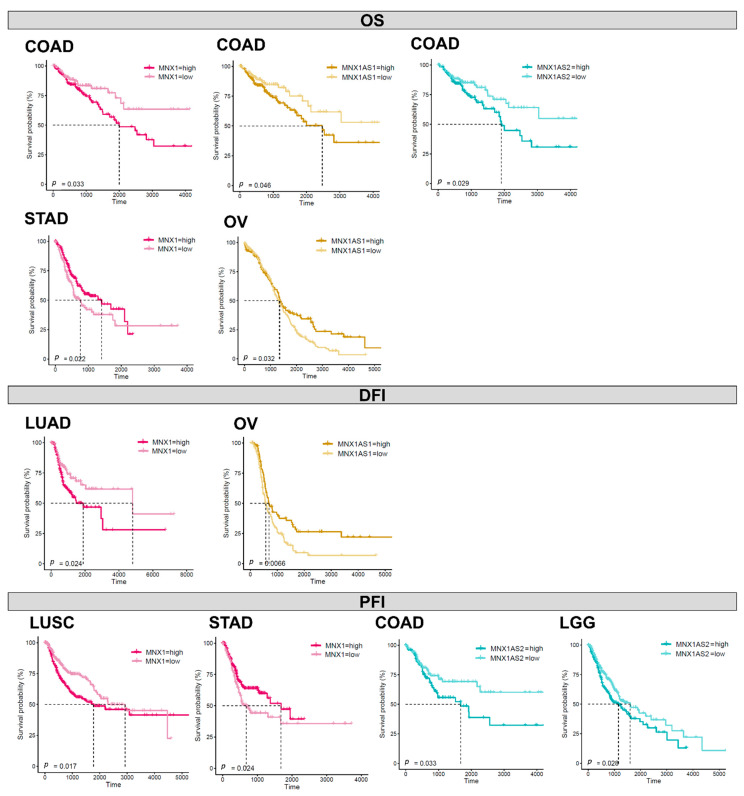
Prognostic evaluation of *MNX1*, *MNX1*-*AS1*, and *MNX1*-*AS2* expressions. Kaplan–Meier plots of statistically significant associations between individual expressions of *MNX1*, *MNX1*-*AS1*, and *MNX1*-*AS2* in OS, DFI, and PFI.

**Figure 8 cells-11-03577-f008:**
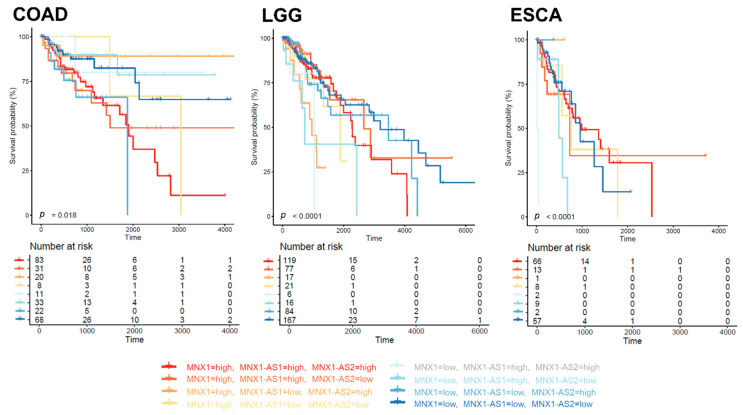
Combinatorial survival analysis of *MNX1*, *MNX1-AS1*, and *MNX1*-*AS2* expressions. Kaplan–Meier plots of OS in COAD, LGG, and ESCA with patients stratified according to expression levels.

**Table 1 cells-11-03577-t001:** Clinical details of TCGA and GTEx cohorts.

Organ	Breast	Colon		Oesophagus	Brain		Liver	Lung		Ovary	Pancreas	Prostate	Skin	Stomach	Testis	Thyroid	Uterus	
TCGA Cancer and Abbreviation	Breast Cancer BRCA	Colon Adenocarcinoma COAD	Rectum Adenocarcinoma READ	Oesophageal Carcinoma ESCA	Glioblastoma Multiforme GBM	Brain Lower Grade Glioma LGG	Liver Hepatocellular Carcinoma LIHC	Lung Adenocarcinoma LUAD	Lung Squamous Cell Carcinoma LUSC	Ovarian Serous Cystadenocarcinoma OV	Pancreatic Adenocarcinoma PAAD	Prostate Adenocarcinoma PRAD	Skin Cutaneous Melanoma SKCM	Stomach Adenocarcinoma STAD	Testicular Germ Cell Tumours TGCT	Thyroid Carcinoma THCA	Uterine Corpus Endometrial Carcinoma UCEC	Uterine Carcinosarcoma UCS
Samples																		
Normal tissue (GTEx)	179	307	307	650	1141	1141	109	288	288	86	167	100	788	173	164	279	77	77
Primary tumour (TCGA)	1212	331	103	195	171	523	421	574	548	427	183	548	102	450	154	571	204	57
Clinical stage																		
Stage I	182 (15.6%)	44 (14.3%)	12 (11.7%)	18 (9.2%)	N/A	N/A	169 (40.1%)	306 (53.3%)	269 (48.7%)	1 (0.2%)	21 (11.5%)	N/A	3 (2.5%)	58 (12.9%)	104 (67.5%)	283 (49.6%)	N/A	22 (38.6%)
Stage II	617 (53.0%)	110 (35.8%)	24 (23.3%)	76 (39.0%)	N/A	N/A	86 (20.4%)	135 (23.5%)	178 (32.2%)	24 (5.6%)	147 (80.3%)	N/A	66 (55.0%)	121 (26.9%)	12 (7.8%)	51 (8.9%)	N/A	5 (8.8%)
Stage III	248 (21.3%)	82 (26.7%)	33 (32.0%)	55 (28.2%)	N/A	N/A	77 (18.3%)	96 (16.7%)	89 (16.1%)	332 (77.2%)	3 (1.6%)	N/A	2 (1.7%)	169 (37.6%)	14 (9.1%)	112 (19.6%)	N/A	20 (35.1%)
Stage IV	20 (1.7%)	40 (13.0%)	13 (12.6%)	9 (4.6%)	N/A	N/A	5 (1.2%)	28 (4.9%)	8 (1.4%)	63 (14.7%)	4 (2.2%)	N/A	27 (22.5%)	41 (9.1%)	0 (0.0%)	55 (9.6%)	N/A	10 (17.5%)
N/A	97 (8.3%)	31 (10.1%)	21 (20.4%)	37 (19.0%)	N/A	N/A	84 (20.0%)	9 (1.6%)	8 (1.4%)	10 (2.3%)	8 (4.4%)	N/A	22 (18.3%)	61 (13.6%)	24 (15.6%)	70 (12.3%)	N/A	0 (0.0%)
Age GTEx (years)																		
Age < 50	64 (35.7%)	108 (35.1%)		252 (38.7%)	178 (15.6%)	343 (68.9%)	29 (26.6%)	92 (32.1%)		38 (44.2%)	67 (40.1%)	44 (44.0%)	266 (33.8%)	72 (41.6%)	62 (37.9%)	86 (30.9%)	38 (49.3%)	
Age > 50	115 (64.3%)	199 (64.9%)		398 (61.3%)	963 (84.4%)	155 (31.1%)	80 (73.3%)	195 (67.9%)		48 (55.8%)	100 (59.9%)	56 (56.0%)	522 (66.2%)	101 (58.4%)	102 (62.1%)	193 (69.1%)	39 (50.6%)	
Age TCGA (years)																		
Age < 50	328 (30.1%)	44 (15.4%)	17 (18.7%)	27 (14.9%)	36 (23.7%)	343 (68.9%)	71 (19.6%)	34 (8.3%)	20 (3.7%)	101 (23.9%)	24 (13.5%)	34 (6.9%)	15 (14.7%)	34 (8.3%)	122 (96.1%)	272 (55.9%)	11 (6.2%)	0 (0.0%)
Age > 50	763 (69.9%)	242 (84.6%)	74 (81.3%)	154 (85.1%)	116 (76.3%)	155 (31.1%)	291 (80.4%)	375 (91.7%)	519 (96.3%)	322 (76.1%)	154 (86.5%)	461 (93.1%)	87 (85.3%)	375 (91.7%)	5 (3.9%)	215 (44.1%)	166 (93.8%)	57 (100.0%)

## Data Availability

Clinical phenotype and expression data is publicly available and freely accessible in the TCGA-TARGET-GTEX cohort from the University of California Santa Cruz public repository Xena accessible online at www.xenabrowser.net (last accessed on 20 October 2022) [22].

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
