# Peer review of "Pan-Cancer Analysis Identifies MNX1 and Associated Antisense Transcripts as Biomarkers for Cancer"

_cells, 2022, doi:10.3390/cells11223577_

Round 1

Reviewer 1 Report

The study shows  that MNX1 and its associated MNX1-AS1 and MNX1-AS2 lncRNA are potential biomarkers in cancer. Several studies showed thta taken individually each RNA can be associated with specific cancer types. Here, authors investigate the clinical relevance of the three RNAs combined ("combo"). To do so, publicly available datasets from TCGA and GTEx are analysed for differential expression (normal vs tumour), clinical relevance, classification and prognosis. The manuscript is well written and studies properly conducted and reported. I however feel that no clear information is gathered from the study.

1) the combo is often behaving in term of scoring as RNAs taken individually. AUC for prediction for instance are quite close for all cancer types. The relevance for an implementation in the clinic is low.

2) the data are correlative. There is no functional analysis of these associations or replication using a different cohort of patients. This must be done for specific cancer types showing the strongest effect of the 'combo' vs individual RNAs.

3) the finding that MNX1 and AS1/2 can classify tumour subtypes must be validated using other transcriptomic data from other groups. this is an intriguing finding that needs replication. It would also be nice to further understand what are these cancer subtypes using the RNA-seq data available (type of pathways ? gene network?) that may be associated with the 'combo' or individual RNAs.

4) PCA is strongly influenced by datasets. Are confounding factors such gender taken into accound in these analyses ? same question for prognostic tests (KM plot) ?

Reviewer 2 Report

The manuscript represents an entirely in silico study, which has been mainly based on multiple TCGA and GTEx data sets, bundled with the help of the Xena platform and a few other tools; with a significant number of different analyses performed with these data - always centered on MNX1 and its 2 companions. The quality of the bioinformatics analysis, and particularly the visualizations and plotting of the correlative data, is always high quality and intuitive; I personally find the approach and figures showing the results rather informative and satisfying. It's also nice to see so many different designs of plots in a single paper - looking at the data from so many different angles. 

Nevertheless, the study would clearly benefit from ANY further, defining studies (even if they are entirely in silico), related to the expression and maybe, also functions of MNX1 and its 2 accessory lncRNAs. I understand that performing any in vitro functional studies is not really matching the scope of the current manuscript. BUT: The authors have almost exclusively looked into the diagnostic value of expression of these 3 genes across most cancer types. But what is the gene and ints 2 sidekicks doing? This is questioned in the introduction, but then the authors don't really engage in answering that question at all. I can understand they want to focus on one area (diagnostic/prognostic value), but if a prognostic gene is also engaged in a highly cancer-relevant function, this often increases the physiological relevance of the findings massively (and the interest of the reader). 

Which genes/gene sets and GO categories, for example, would be correlating with high/low MNX1 expression? Are there any statistically significant enrichments that could point towards potential functions, in cancer-relevant processes? Is expression mainly found in the tumor cells, or also stromal, endothelial, or even immune cells, within the tumors? 

other issues: the tumor types examined in the paper are not introduced properly. Instead, only the abbreviations (UCEC, PRAD, LUAD, ESCA, and COAD/READ, LIHC, STAD, TGCT, and THCA established by TCGA are used here, and I think the need introduction in the text for all those readers not familiar with the types. I find papers hard to read that are just full with unexplained abbreviations; and nobody goes to the list of abbreviations to check all of them out. 

As introduced in Fig. 1 C and A, the antisense-ncRNA MNX_AS1 may simply be under the same transcriptional control as the "mother" gene MNX1. IN contrast, MNX1_AS2 is clearly independently expressed and controlled, as clearly and potently shown in Fig. 4.  That observation could be quite simply explored by low-angle bioinformatics, to clarify who and what controls the promoter regions? 

Round 2

Reviewer 1 Report

I found the newest version of the manuscript improved and all my questions have been answered. I noticed a few typos in the revised version (e.g. lane 481).